# Serum Bile Acids Concentrations and Liver Enzyme Activities after Low-Dose Trilostane in Dogs with Hyperadrenocorticism

**DOI:** 10.3390/ani13203244

**Published:** 2023-10-18

**Authors:** Nannicha Tinted, Smith Pongcharoenwanit, Thodsapol Ongvisespaibool, Veerada Wachirodom, Taksaon Jumnansilp, Narinthip Buckland, Piyathip Chuchalermporn, Sirikul Soontararak, Selapoom Pairor, Jörg M. Steiner, Naris Thengchaisri, Sathidpak Nantasanti Assawarachan

**Affiliations:** 1Internal Medicine Unit, Kasetsart University Veterinary Teaching Hospital, Faculty of Veterinary Medicine, Kasetsart University, 50 Ngamwongwan Rd., Lat Yao, Jatujak, Bangkok 10900, Thailand; nannicha.tin@ku.th; 2Department of Companion Animal Clinical Sciences, Faculty of Veterinary Medicine, Kasetsart University, 50 Ngamwongwan Rd., Lat Yao, Jatujak, Bangkok 10900, Thailand; sirikul.s@ku.th (S.S.); selapoom_ake@hotmail.com (S.P.); ajnaris@yahoo.com (N.T.); 3Endocrinology and Gastroenterology Unit, Kasetsart University Veterinary Teaching Hospital, Faculty of Veterinary Medicine, Kasetsart University, 50 Ngamwongwan Rd., Lat Yao, Jatujak, Bangkok 10900, Thailand; smith.po@ku.th (S.P.); thodsapol.o@ku.th (T.O.); veerada.w@ku.th (V.W.); taksaon.j@ku.th (T.J.); narinthip.bu@ku.th (N.B.); 4Radiology Unit, Kasetsart University Veterinary Teaching Hospital, Faculty of Veterinary Medicine, Kasetsart University, 50 Ngamwongwan Rd., Lat Yao, Jatujak, Bangkok 10900, Thailand; piyathip.c@ku.th; 5Gastrointestinal Laboratory, Department of Small Animal Clinical Sciences, School of Veterinary Medicine and Biomedical Sciences, Texas A&M University, College Station, TX 77843, USA; jsteiner@cvm.tamu.edu

**Keywords:** dogs, hyperadrenocorticism, liver function, serum bile acids, hepatopathy

## Abstract

**Simple Summary:**

Steroid-induced hepatopathy is the most important liver abnormality in dogs with hyperadrenocorticism (HAC). Serum activities of several liver enzymes increase in dogs with HAC. However, serum bile acids concentrations in dogs with HAC are poorly described. This study evaluated serum total bile acids concentrations and serum liver enzyme activities after trilostane treatment in dogs with HAC.

**Abstract:**

Hyperadrenocorticism (HAC) often leads to vacuolar hepatopathy. The impact of trilostane treatment on serum total bile acids (SBAs) concentrations in dogs with HAC remains unknown. This study investigated SBAs concentrations in healthy dogs and those with HAC following trilostane therapy. Ten healthy dogs and fifteen dogs with HAC were prospectively enrolled. A biochemistry profile and pre- and post-prandial SBAs concentrations were determined in each dog. Dogs with HAC were reassessed at 1 and 3 months after the initiation of trilostane treatment. Dogs with HAC had significantly higher serum ALT, ALP, and GGT activities, and cholesterol, triglyceride, and pre-prandial SBAs concentrations compared to healthy dogs. After 3 months of trilostane treatment, polyuria/polydipsia and polyphagia were completely resolved in 42.8% and 35.7%, respectively. Significant improvements in serum ALT and ALP activities and cholesterol concentrations were observed within 1–3 months of trilostane treatment. However, pre- and post-prandial SBAs concentrations did not significantly decrease. These findings suggest that treatment with low-dose trilostane for 3 months appears to reduce serum liver enzyme activities, but not SBAs concentrations. Further investigation is warranted to explore the effects of low-dose trilostane treatment on SBAs concentrations for a longer duration or after achieving appropriate post-ACTH cortisol levels.

## 1. Introduction

Hyperadrenocorticism (HAC) is one of the most common endocrinopathies of middle-aged and older dogs. It results from the excessive production of cortisol by the adrenal cortex [1,2]. It is estimated that approximately 85% of dogs with HAC have the pituitary-dependent form (PDH), while 15% have the adrenal-dependent form (ADH) [3,4]. The observed clinical signs in dogs with HAC reflect the gluconeogenic, lipolytic, protein catabolic, anti-inflammatory, and immunosuppressive effects of excess glucocorticoid hormones [5]. The most common clinical signs reported in dogs with HAC are polyuria, polydipsia, polyphagia, panting, abdominal distension, and endocrine alopecia [4,5,6]. Common hematological findings reported in dogs with HAC include neutrophilia, eosinopenia, lymphopenia, monocytosis, and erythrocytosis. Common biochemical findings reported in dogs with HAC include increased alkaline phosphatase (ALP) and alanine aminotransferase (ALT) activities, hypercholesterolemia, hypertriglyceridemia, decreased urea concentration, and hyperglycemia [5]. The purpose of treatment is to eliminate the clinical signs caused by chronic glucocorticoid excess. Treatment options for HAC are medical and surgical management, depending on the cause of the disease and patient condition [4]. Previously, the most common medical treatment for dogs with HAC was mitotane, an adrenocorticolytic drug [7]. Since 2000, trilostane has been increasingly used in dogs with HAC. Trilostane is a synthetic hormonally inactive steroid that competitively inhibits 3β-hydroxysteroid dehydrogenase, resulting in the suppression of the synthesis of cortisol produced autonomously within an adrenocortical tumor or in response to adrenocorticotropic hormone (ACTH) stimulation [1,7,8,9,10]. 

Vacuolar hepatopathy is one of the most common histological diagnosis in dogs [11,12,13,14,15]. High concentrations of circulating corticosteroids lead to severe vacuolar or steroid-induced hepatopathy [16]. Steroid-induced hepatopathy is characterized by vacuolar change and swelling of hepatocytes, resulting from glycogen accumulation [17]. In addition, affected dogs usually have increased serum ALP, gamma-glutamyl transpeptidase (GGT) activities, and cholesterol concentrations. Significant ultrasonographic changes in the liver due to vacuolar hepatopathy may also be found [18]. Vacuolar hepatopathy should be taken seriously as severe vacuolar hepatopathy may not always be reversible and may progress to fulminant hepatic dysfunction [13]. 

Serum total bile acids (SBAs) concentrations are often used to assess hepatic function in non-icteric dogs [19,20]. Increased SBAs concentrations have been demonstrated in dogs with reduced liver function, portosystemic shunting, or cholestasis [20,21,22,23]. Bile acids are synthesized in the liver from cholesterol [20,24] and, immediately after synthesis, are secreted into bile and concentrated for storage in the gallbladder. Eating stimulates cholecystokinin (CCK) release, which causes gallbladder contraction, which in turn releases bile into the duodenum through the sphincter of Oddi. Most bile acids contained in the bile are reabsorbed in the ileum and are transported back to the liver via the portal blood circulation [21]. Typically, SBAs are measured after withholding food for 12 h (pre-prandial SBAs) and 2 h after a small high-fat meal (post-prandial SBAs) [19,20]. Measurement of post-prandial SBAs is a more sensitive marker than resting SBAs concentrations for the diagnosis of liver disease [23,25]. Elevated pre- and post-prandial SBAs concentrations have been documented in a substantial proportion of dogs with HAC, with rates reported at 11.8% and 37.5%, respectively [26]. However, the underlying cause of these increased SBAs concentrations in dogs with HAC remains unknown. Importantly, there is limited information regarding SBAs concentrations in dogs with HAC subsequent to treatment with trilostane.

The objectives of this study were to evaluate liver enzyme activities (i.e., ALT, ALP), lipid profiles (i.e., cholesterol, triglycerides), and SBAs concentrations (both pre- and post-prandial values) in healthy dogs and dogs with HAC before and after treatment with trilostane.

## 2. Materials and Methods

In total, 15 client-owned dogs newly diagnosed with spontaneous HAC and 10 healthy control dogs were prospectively enrolled at the Kasetsart University Veterinary Teaching Hospital, Bangkok, Thailand. Kasetsart University Institutional Animal Care and Use Committee approval (ACKU65-VET-039) and owner consent were obtained. Dogs were considered healthy if no abnormalities were detected based on their history and physical examination at the time of blood collection as well as the preceding 6 months. The results of a complete blood count (CBC), serum concentrations of blood urea nitrogen (BUN), creatinine, total bilirubin, total protein, albumin, basal cortisol, and blood glucose and serum activities of ALT, aspartate aminotransferase (AST), ALP, and GGT of the healthy control dogs were also within the respective reference intervals. Healthy control dogs had unremarkable findings on complete abdominal ultrasound. In dogs with HAC, diagnosis was made based on history, clinical signs, physical examination, routine blood testing, endocrine function testing, and ultrasonography. Abdominal ultrasound was performed by the Thai Board of Veterinary Surgeons, sub-specialty of veterinary diagnostic imaging, using a real-time scanner (LOGIQ E9, GE, Fairfield, CT, USA) with a 13 MHz broadband linear transducer. The endocrine tests used to make the diagnosis of HAC included an ACTH stimulation test and a low-dose dexamethasone suppression test (LDDST). The diagnosis was confirmed if the results of LDDST or ACTH stimulation testing were consistent with a diagnosis of HAC, along with compatible clinical signs [9]. In all cases, the differentiation of PDH was based on the ultrasonographic appearance of the adrenal glands and additionally, in some cases, on the results of the LDDST [27]. The presence of symmetrical and bilateral adrenal gland enlargement, indicated by an adrenal gland thickness greater than 7 mm, was considered indicative of PDH [1]. In all dogs, there were no changes in dietary type from 3 months before starting until the end of the study. In addition, hepatoprotectants and choleretic drugs that had been given before the diagnosis of HAC were not changed throughout the study. 

Dogs with HAC were assessed before the initiation of treatment with trilostane (pre-treatment) and 1 month and 3 months after treatment with trilostane, while healthy control dogs presented once. Food was withheld from all dogs for at least 12 h before the collection of blood samples. The CBC was completed using an automated hematology analyzer (Sysmex XN-1000TM Hematology Analyzer; Sysmex; Mundelein, IL, USA). A serum biochemical profile was performed using an automatic chemistry analyzer (IL Lab 650 Chemistry System; Diamond Diagnostics; Holliston, MA, USA) and included the measurement of serum activities of ALT, AST, ALP, GGT, concentrations of BUN, creatinine, total bilirubin, triglyceride, cholesterol, total protein, albumin, sodium, and potassium. Blood glucose concentration was measured using an Alphatrak 2 glucometer (Zoetis; Bangkok, Thailand). SBAs concentration was measured using a Catalyst One Chemistry Analyzer (IDEXX Laboratories, Inc.; Westbrook, ME, USA). Once pre-prandial (fasting) blood samples had been collected, the dogs were fed Hill’s^®^ Prescription Diet^®^ a/d^®^ Canine/Feline (10 g/kg body weight) [28]. Post-prandial SBAs concentrations were obtained at 120 min after the meal. 

Trilostane (Vetoryl^®^, Dechra, Kansas, KS, USA) was administered orally at an initial dose of 1 mg/kg of body weight, once daily in the morning; this regimen was maintained for 4 weeks. The ACTH stimulation test was reassessed after 1 and 3 months. The ACTH stimulation test was undertaken 4–6 h after trilostane administration. Serum cortisol concentrations were measured before and 1 h after the intravenous administration of synthetic ACTH (5 µg/kg; Synacthen^®^ [Atnahs Pharma UK Ltd.; Basildon, UK]). Cortisol concentration was measured using a chemiluminescent immunoassay (Immulite 1000^®^; Siemens Healthcare Diagnostics Inc.; Flanders, NJ, USA). The range of post-ACTH cortisol concentration considered indicative of good cortisol control was 2.0–5.0 µg/dL [27]. Trilostane dosage was adjusted individually based on clinical signs and post-ACTH cortisol concentration. If the dog showed an improvement in clinical signs and an ACTH-stimulated serum cortisol concentration between 2.0–5.0 µg/dL, the trilostane dose was maintained. In dogs responding well to treatment but with a high ACTH-stimulated cortisol concentration, the trilostane dose was also left unchanged [3]. 

Data analyses were performed using GraphPad Prism version 10.0.2 software (GraphPad Software, Inc.; La Jolla, CA, USA). The Shapiro–Wilk test was used to assess the normality of continuous variables. The statistical significance of differences was evaluated within the group concerning laboratory analytes during treatment using either repeated measures ANOVA or the Friedman test with a Dunn’s multiple comparison post hoc analysis. Differences in biochemical variables between healthy and HAC dogs were tested based on Student’s *t*-test or the Mann–Whitney U test depending on the normality of the data. For statistical analyses, if the cortisol or SBAs concentration were less than 1, their value was set at 0.9. Since not all data followed a normal distribution, continuous variables were presented as the median and range. The significance level was set as *p* < 0.05.

## 3. Results

The sex, size, and median (range) age, body weight, and body condition scores (BCS) of both groups are summarized in Table 1. Breeds represented in the control group were Pomeranians (*n* = 6), Samoyeds (*n* = 2), Shiba Inu (*n* = 1), and a mixed-breed (*n* = 1). The HAC group included Shih Tzus (*n* = 5), mixed-breeds (*n* = 3), Beagles (*n* = 2), Pomeranians (*n* = 2), Poodles (*n* = 2), and a Chihuahua (*n* = 1). Dogs with HAC had significantly higher BCS and ages than healthy control dogs (*p* < 0.05).

In total, there were twelve dogs with PDH (80%) and three with ADH (20%). Dogs with HAC were initially treated with low-dose trilostane (1.25 ± 0.3 mg/kg) administered once daily. Three months later, the trilostane dose was adjusted to 1.1 ± 0.8 mg/kg every 24 h. Dogs diagnosed with HAC exhibited various clinical signs, including polyuria (14/15 dogs), polydipsia (14/15 dogs), polyphagia (14/15 dogs), abdominal distension (13/15 dogs), and alopecia (9/15 dogs) (Table 2). Following 1 month of treatment, the clinical signs of polyuria and polydipsia had improved in eight of fourteen (57.1%) dogs and had completely resolved in three out of fourteen (21.4%) dogs. After 3 months of treatment, polyuria and polydipsia had improved in seven out of fourteen (50%) dogs and had completely resolved in six out of fourteen (42.8%) dogs. Among the fourteen dogs with polyphagia, seven (50%) dogs showed improvement after 1 month of treatment and in three (21.4%) dogs, polyphagia had completely resolved. After 3 months of treatment, seven (50%) dogs demonstrated an improvement in polyphagia, and in five (35.7%) dogs polyphagia had completely resolved.

The median (range) cortisol concentrations after ACTH stimulation were 42 (13.3 to 50), 10.4 (3.8 to 32.5), and 9.2 (4.7 to 24.1) µg/dL at 0, 1, and 3 months, respectively (Table 3). At each re-evaluation, the post-ACTH cortisol concentration was significantly lower than the concentration at the start of the study (*p* < 0.05). At enrollment, all the dogs had ALP activities above the upper limit of the reference interval; 86.7% (13/15) had increased ALT activities; and 66.7% (10/15) had increased GGT activities. Also, 73.3% (11/15) had hypertriglyceridemia and 53.3% (8/15) had hypercholesterolemia at time of enrollment. Serum ALT and ALP activity and cholesterol and triglyceride concentrations in HAC patients were significantly higher than those in healthy dogs (*p* < 0.05). Serum activities of ALT and ALP and serum concentrations of cholesterol at 1 and 3 months were significantly lower than the values before treatment (*p* < 0.05) (Table 3). However, serum ALP activity in all dogs remained higher than the upper limit of the reference interval by the end of the study. After 3 months of trilostane treatment, ten (76.9%) and six (75%) dogs demonstrated increased serum ALT activities and increased serum cholesterol concentrations, respectively. In addition, the serum triglyceride concentrations of dogs with HAC did not significantly differ before and after treatment. Concentrations of bilirubin and glucose in both healthy dogs and dogs with HAC were within the respective reference intervals and there were no significant differences between the groups, both before and after trilostane treatment. Throughout the study, there were no significant changes observed in sodium and potassium concentrations (*p* = 0.49 and 0.15). After 1 and 3 months of treatment with trilostane, seven dogs presented with hyperkalemia. Additionally, a ratio of serum sodium-to-potassium less than 27 was found in five and three dogs after 1 month and 3 months of trilostane treatment, respectively.

Pre-prandial SBAs concentrations of the HAC dogs at baseline (median [range]; 20.6 [1.1 to 118]) were significantly higher than those of the healthy control dogs (median [range]; 0.9 [0.9 to 3.6]) (Table 3). However, post-prandial SBAs concentrations at baseline were not significantly different between healthy dogs and dogs with HAC. There were no significant changes in the pre-prandial or post-prandial SBAs concentrations of dogs with HAC during the treatment. Abnormal pre- (>14.9 µmol/L) and post-prandial (>29.9 µmol/L) SBAs concentrations were observed in 53.3% (8/15) and 40% (6/15), respectively, of dogs with HAC. After 3 months of treatment, the resolution of abnormally high pre-and post-prandial SBAs concentrations was identified in four out of eight (50%) and one out of six (16.7%) dogs, respectively. As expected, in healthy dogs, post-prandial SBAs concentrations were consistently higher than pre-prandial SBAs concentrations (Figure 1A). However, HAC dogs before treatment showed a different pattern, with many dogs having post-prandial SBAs concentrations that were indistinguishable from pre-prandial SBAs concentrations (Figure 1B). After 1 and 3 months of trilostane treatment, their patterns of pre- and post-prandial SBAs were more similar to those of healthy dogs (Figure 1C,D).

## 4. Discussion

This study compared SBAs concentrations and serum liver enzyme activities in dogs with HAC to those in healthy control dogs. HAC dogs were enrolled before the initiation of therapy and then treated with trilostane and followed up after 1 and 3 months of trilostane therapy. Dogs with HAC had significantly higher pre-prandial SBAs concentrations than healthy dogs. However, only half of the dogs with HAC demonstrated increased pre-prandial SBAs, and these increases were minor. The increase in pre-prandial SBAs in dogs with HAC may not be clinically relevant. A previous study reported increased pre-prandial SBAs in only 11.8% (2/17 dogs) of dogs with HAC [26]. It is possible that the increased pre-prandial SBAs observed in dogs with HAC are the result of some degree of cholestasis, but hepatic dysfunction cannot be ruled out. The other liver function tests, including albumin, bilirubin, BUN, and glucose, were within their respective reference limits in these dogs, further supporting a lack of hepatic dysfunction. It has been proposed that cholestasis in dogs with steroid-induced hepatopathy may occur due to the compression of bile canaliculi [29]. Because of their emulsifying action, bile acids can be toxic to the epithelial cell membranes lining the biliary system and to hepatocytes [21,30]. During a cholestatic state, hepatocytes use a mechanism to decrease the toxicity of bile acids by reversing certain sinusoidal transporters which pump bile acids into the circulation rather than into the canaliculus [21]. Another explanation of high pre-prandial SBAs in dogs with HAC may be due to a paradoxical SBAs phenomenon, whereby pre-prandial SBAs exceeds the post-prandial SBAs concentration [19,31]. Traditionally, such paradoxical results have been attributed to dogs seeing or smelling food, but this has been disproven [19]. Instead, it is believed that gallbladder motility is the underlying cause for such paradoxical results. Gallbladder motility in turn is related to CCK release, which may be regulated, in part, by the migrating motor complex (MMC), so that a rhythmically appearing MMC may lead to CCK release and gallbladder contraction independent of food uptake [32,33]. Three dogs with HAC in our study demonstrated paradoxical SBAs concentrations. The proposed underlying causes of this phenomenon include spontaneous gallbladder contraction during fasting, differences in the emptying rate of gastric contents, variations in the rate and efficacy of absorption, and the timing and degree of CCK release [19,20,21]. 

After treatment once daily with a low dose of trilostane, there was a declining trend in pre-prandial SBAs concentrations, although no significant changes were demonstrated. The number of dogs with increased pre-prandial SBAs (>14.9 µmol/L) fell from eight dogs to four dogs after 3 months of treatment. We proposed that the treatment of HAC dogs with trilostane may have the potential to alleviate bile retention and/or liver dysfunction in HAC dogs. 

One study reported a higher sensitivity of post-prandial SBAs for the diagnosis of liver disease than that of pre-prandial SBAs [23]. Our study showed no significant differences in post-prandial SBAs concentrations between dogs with HAC and healthy control dogs. Forty percent of dogs with HAC had increased post-prandial SBAs, similar to the result from a previous study (37.5%; 6/16 dogs) [26]. Post-prandial SBAs is influenced by the responsiveness of gallbladder contraction after a meal and by the efficiency of bile acids being removed from the blood stream by hepatocytes [23,29]. Dogs with HAC may present with gallbladder dysmotility that could affect post-prandial SBAs. This hypothesis of gallbladder dysmotility is supported by a report which states that dogs with HAC had 29.0 times higher an incidence of gallbladder mucocele than dogs without HAC [34]. One of the predisposing factors to gallbladder mucocele development is gallbladder dysmotility. Other studies found that steroid hormones, specifically progesterone, testosterone, and dihydrotestosterone, resulted in a concentration-dependent inhibition of gallbladder motility in guinea pigs [35,36]. Hypercortisolemia in dogs with HAC may disrupt the normal motility of the gallbladder, leading to the development of a gallbladder mucocele [32]. In the present study, the overall pattern of pre- and post-prandial SBAs concentrations in HAC dogs after treatment appears to be more similar to healthy dogs. The reason for this normalization is not entirely clear. However, one possible explanation is that gallbladder motility is compromised in dogs with HAC and treatment with trilostane helps to normalize gallbladder motility. Further study of gallbladder motility in dogs with HAC should be considered. 

Dogs with HAC exhibited significantly higher serum ALT, ALP, and GGT activities compared to healthy control dogs. Elevated ALT activity, a marker for liver injury, may result from corticosteroid-induced hepatic cell injury or vacuolar hepatopathy. Increased ALP activity, a marker for cholestasis, is due to a combination of the increased synthesis of the steroid-induced ALP isoenzyme and vacuolar hepatopathy [13,29,37]. Upon treatment with a once-daily low dose of trilostane (1.25 ± 0.3 mg/kg every 24 h), a significant decrease in serum ALT and ALP activities was observed within 1 month of treatment. These findings aligned with the significant reduction in ACTH-stimulated cortisol concentration. Our outcomes were consistent with prior studies that reported a significant decrease in serum ALT and/or ALP activity. However, wide variations in trilostane dosages were used in previous studies [1,3,38,39]. Two studies described a decrease in serum ALT and/or ALP activity in dogs with HAC after 1 month of high-dose trilostane treatment (2.4–15 mg/kg/day [1] and 30 mg/day for a dog weighing less than 5 kg [3]). Another study reported significantly lower ALP and ALT activities at 12 and 24 weeks of low-dose trilostane treatment (0.78 ± 0.26 mg/kg every 12 h), respectively [3]. It is known that increased serum ALP activities in dogs with HAC is predominantly caused by the corticosteroid isoenzyme [29]. The decrease in ALT and ALP activities in dogs with HAC after trilostane treatment is thus suggested to be a direct result of lowered serum glucocorticoid accumulation. In addition, we propose that the reduction in serum glucocorticoid may lead to reduced glycogen accumulation and cell swelling of hepatocytes, leading to a decrease in cholestasis. The serum ALP activity of all dogs with HAC in this study was still above the upper limit of the reference interval after treatment with trilostane. These findings could have been due to the short duration of low-dose trilostane treatment and unsatisfactory post-ACTH cortisol levels in the present study. Enzymatic induction due to the accumulation of other cortisol precursors and concurrent disease processes were also possible [39,40]. All dogs with HAC were supplemented with hepatoprotectants and choleretic drugs for an average duration of 4.2 months before the diagnosis of HAC was confirmed. However, there were no significant reductions in serum ALT and ALP activity after the administration of hepatic supplements (Appendix A). 

Dogs with HAC had significantly higher serum cholesterol and triglyceride concentrations compared to healthy dogs. After treatment with low-dose trilostane (1.25 ± 0.3 mg/kg), the concentration of cholesterol (but not that of triglyceride) decreased significantly at 1 month after re-evaluation. A previous study has demonstrated a significant decrease in serum cholesterol concentrations after 1 month of high-dose trilostane (30 mg once daily for a dog weighing less than 5 kg) and after 6 months of low-dose trilostane (0.78 ± 0.26 mg/kg, twice daily) [3]. Hyperlipidemia in dogs with HAC is attributed to the effect of cortisol, which enhances the synthesis of hormone-sensitive lipase, leading to an increase in free fatty acids entering the circulation [41]. Additionally, cortisol reduces the sensitivity of the anterior pituitary to thyrotropin-releasing hormone, subsequently decreasing thyroid-stimulating hormone output and resulting in a relative hypothyroid state [42]. Improvement in the lipid enzymatic pathway, resulting from decreased serum cortisol concentrations, may account for the reduction in serum cholesterol concentrations with trilostane therapy [3]. In addition, a previous study of dogs with chronic hepatitis demonstrated that there were positive associations between hyperlipidemia, especially hypercholesterolemia, and serum ALT, ALP, and GGT activity and SBAs concentration [42]. Hypercholesterolemia in dogs with HAC may be correlated with the severity of liver injury and cholestasis. Thus, the underlying rationale for the amelioration of hypercholesterolemia in dogs with HAC after treatment may be related to the improvement in liver injury and cholestasis. 

Dogs with HAC manifested various clinical signs. Approximately 43% (6/14) of dogs demonstrated a complete resolution of polyuria and polydipsia after 3 months of treatment. Although lower-than-recommended dosages of trilostane and inappropriate levels of post-ACTH cortisol were evident in this study, most dogs showed improvements in terms of clinical signs and serum biochemical profile abnormalities, namely ALT and ALP activity and cholesterol concentration within 1 and 3 months of initiating therapy. Our results were similar to another study which reported 44.4% (4/9) of dogs with normal water intake at 3 months after treatment using twice-daily, low-dose trilostane [3]. The dose of trilostane recommended by the manufacturer (Dechra, KS, USA) is 2.2–6.7 mg/kg once daily. However, a lower dose of trilostane has been proven to be effective to control the disease [27,43,44]. The variations in dosage and frequency of trilostane administration in different studies could have led to differing results and such differences should be carefully considered when interpreting the findings. Future studies may be needed to validate and better understand the implications of these results. 

Reported adverse effects of trilostane include anorexia, lethargy, vomiting, diarrhea, hyperkalemia, and hyponatremia [43]. The inhibition of steroid production through trilostane treatment can lead to hypoadrenocorticism in dogs. An abnormal serum sodium-to-potassium ratio may indicate hypoadrenocorticism in HAC dogs treated with trilostane [3]. The assessment of the adrenal reserve during treatment can be accomplished through an ACTH stimulation test [38]. In the current study, one-half of our dogs showed mild hyperkalemia without clinical signs and a decreased sodium-to-potassium ratio. Notably, in these dogs, ACTH stimulation testing did not indicate hypoadrenocorticism. The mechanism behind the mild hyperkalemia in dogs treated with trilostane remains unknown [43]. It is possible that trilostane may have a greater impact on blocking the synthesis of mineralocorticoids than glucocorticoids [43,44]. In addition, a histopathological study of the adrenal glands of dogs that had received trilostane evidenced adrenocortical necrosis [45]. This could potentially explain the observed electrolyte imbalances in some dogs undergoing trilostane treatment. In contrast, another study demonstrated that treatment with trilostane did not cause a clinically relevant alteration in serum potassium concentrations [46]. Further research is needed to fully understand the underlying mechanisms responsible for these effects.

Our study had several limitations. First, it lacked an age- and sex-matched control group [47]. Additionally, certain dogs with HAC in our study were on a low-fat diet, which could potentially have influenced serum concentrations of cholesterol and triglycerides [48]. Nevertheless, dietary type in all dogs remained the same for at least 3 months preceding the initiation of trilostane. Also, our study population consisted of specific dog breeds, including Beagles (13.3%), that have been reported to be predisposed to hereditary hyperlipidemia [48]. Due to ethical issues, an untreated control group could not be included in this study. Thus, we could not definitively attribute the observed improvements to the therapy. Moreover, two healthy dogs (20%) demonstrated a mild increase in post-prandial SBAs concentration in the present study. Neither of these two dogs had any abnormalities identified upon physical examination, complete blood count, biochemistry profile, or abdominal ultrasonography, but a liver biopsy was not performed in these healthy dogs. Furthermore, most of the dogs in the present study were not clinically controlled, which may be due to the short duration of the study. Future avenues of investigation could involve exploring the effects of trilostane treatment for longer periods of time or at a higher dose, coupled with a re-evaluation of the dogs when the post-ACTH cortisol concentrations align with targeted ranges. These findings would provide insights into the effects of trilostane treatment and its dynamic impact on bile acids concentrations in dogs with HAC.

## 5. Conclusions

Our study showed increased pre-prandial SBAs concentrations in dogs with HAC, although these increases may not be clinically significant. While once-daily administration of low-dose trilostane effectively reduced the serum activities of the liver enzymes ALT and ALP and serum concentrations of cholesterol within the first month of treatment, it did not lead to a substantial improvement in pre-prandial SBAs concentrations. This suggests that trilostane holds promise in mitigating certain aspects of liver injury and cholestasis associated with HAC. Consequently, further research exploring various trilostane dosages and longer treatment durations is warranted to elucidate the intricate relationship between trilostane treatment and SBAs concentration in dogs with HAC. 

## Figures and Tables

**Figure 1 animals-13-03244-f001:**
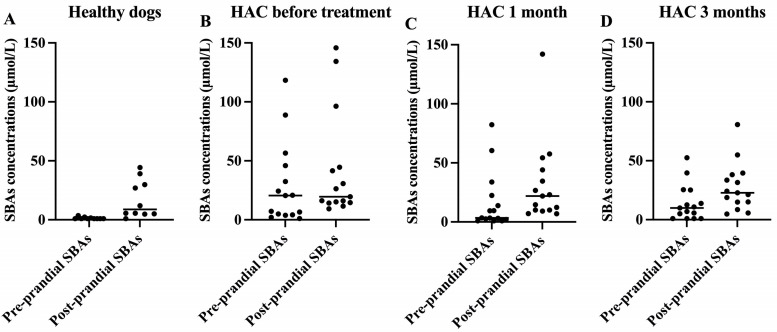
Scatterplots of pre- and post-prandial serum total bile acids (SBAs) concentrations in healthy dogs (**A**), dogs with HAC before treatment (**B**), and dogs with HAC after treatment for 1 month (**C**) and 3 months (**D**). Lines represent medians.

**Table 1 animals-13-03244-t001:** General characteristics of healthy dogs and dogs with HAC.

Parameter	Healthy Dogs	Dogs with HAC
N	10	15
Bodyweight (kg)	4.5 (3.1 to 25.5)	8.7 (3.3 to 30.5)
Nine-point BCS	4 (4 to 6)	7 (6 to 9) *
Age (years)	2 (1 to 3.3)	11.6 (7.1 to 14.5) *
Sex (N)		
Male	4 (4 intact)	7 (3 intact, 4 neutered)
Female	6 (5 intact, 1 spayed)	8 (4 intact, 4 spayed)
Size (N)		
Small (0–12 kg)	7	11
Medium (12.01–24 kg)	1	3
Large (>24.01 kg)	2	1

Results are expressed as median (range). BCS, body condition scores. * Significant difference between healthy and HAC groups (*p* < 0.05).

**Table 2 animals-13-03244-t002:** Clinical signs of dogs with HAC before and 1 and 3 months after treatment with trilostane.

Clinical Signs	Time of Evaluation	No. of Dogs	Improvement N (%)	Complete Resolution N (%)
Polyuria	Pre-treatment	14	-	-
	1 month	-	8 (57.1%)	3 (21.4%)
	3 months	-	7 (50%)	6 (42.8%)
Polydipsia	Pre-treatment	14	-	-
	1 month	-	8 (57.1%)	3 (21.4%)
	3 months	-	7 (50%)	6 (42.8%)
Polyphagia	Pre-treatment	14	-	-
	1 month	-	7 (50%)	3 (21.4%)
	3 months	-	7 (50%)	5 (35.7%)

**Table 3 animals-13-03244-t003:** Biochemical profiles and ACTH-stimulated serum cortisol concentrations in dogs with HAC before and after treatment with trilostane.

Parameter (Reference Range)	Healthy Dogs	Dogs with HAC		
		Pre-Treatment	1 Month	3 Months
Pre-ACTH cortisol level (0.7–9 µg/dL)	5.6 (2.3 to 7.6) ^ab^	4.6 (2.2 to 13.6)	3.9 (1 to 10.3) ^a^	3.6 (1 to 12.5) ^b^
Post-ACTH cortisol level (6–19 µg/dL)	-	42 (13.3 to 50) ^ab^	10.4 (3.8 to 32.5) ^a^	9.2 (4.7 to 24.1) ^b^
ALT (6–70 U/L)	53.5 (35 to 73) ^cde^	239 (27 to 937) ^abc^	127 (20 to 653) ^ad^	127 (18 to 393) ^be^
AST (10–43 U/L)	35.5 (23 to 67)	36 (18 to 143)	29 (18 to 88)	27 (2 to 62)
ALP (8–76 U/L)	58.5 (29 to 86) ^cde^	2049 (303 to 8438) ^abc^	1535 (218 to 5415) ^ad^	985 (150 to 5221) ^be^
GGT (0–15 U/L)	4 (0 to 6) ^abc^	39 (6 to 637) ^a^	21 (5 to 534) ^b^	17 (6 to 402) ^c^
Cholesterol (124–335 mg/dL)	254.5 (179 to 330) ^cde^	352 (239 to 792) ^abc^	300 (144 to 544) ^ad^	331 (166 to 679) ^be^
Triglyceride (26–108 mg/dL)	47 (30 to 98) ^bcd^	171 (79 to 537) ^b^	173 (67 to 477) ^ac^	131 (69 to 259) ^ad^
Total bilirubin (0.1–0.5 mg/dL)	0.1 (0 to 0.2)	0.1 (0 to 0.3)	0.1 (0 to 0.2)	0.1 (0 to 0.2)
Pre-prandial SBAs (0–14.9 µmol/L)	0.9 (0.9 to 3.6) ^abc^	20.6 (1.1 to 118) ^a^	3.4 (0.9 to 82.3) ^b^	9.9 (0.9 to 52.6) ^c^
Post-prandial SBAs (0–29.9 µmol/L)	8.8 (0.9 to 44.3)	19.6 (9.3 to 145.8)	21.9 (6.9 to 42.1)	22.8 (4.8 to 80.7)
BUN (10–26 mg/dL)	22 (12 to 33) ^a^	15 (7 to 36) ^a^	19 (7 to 47)	16 (6 to 61)
Albumin (2.3–3.2 mg/dL)	3.6 (3.1 to 4.1) ^abc^	3.2 (2.8 to 4) ^a^	3.2 (2.8 to 4.1) ^b^	3.2 (2.7 to 3.8) ^c^
Glucose (60–130 mg/dL)	111 (84 to 148)	107 (86 to 169)	106 (66 to 152)	125 (82 to 166)
Na (138–152 mEq/L)	142.3 (137.9 to 148.6)	143.9 (137.5 to 150.5)	143.6 (138.3 to 145.8)	143.1 (140 to 148.2)
K (3.5–5.1 mEq/L)	4.1 (3.6 to 4.8) ^abc^	5 (4.3 to 6) ^a^	5 (4.6 to 6.3) ^b^	5.1 (4.5 to 5.6) ^c^
Na:K ratio (27:1–40:1)	34.9 (29.7 to 38.6) ^abc^	28.6 (23.5 to 33.5) ^a^	28.3 (22.9 to 32) ^b^	28.8 (24.8 to 32.4) ^c^

Results are expressed as medians (range). ALT, alanine aminotransferase; AST, aspartate aminotransferase; ALP, alkaline phosphatase; GGT, gamma-glutamyl transferase; SBAs, serum bile acids; BUN, blood urea nitrogen; Na, sodium; K, potassium. For each variable, the same lowercase superscript indicates statistical significance (*p* < 0.05).

## Data Availability

The datasets generated and/or analyzed during the current study are available from the corresponding author upon reasonable request.

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
