# Peer review of "Serum Bile Acids Concentrations and Liver Enzyme Activities after Low-Dose Trilostane in Dogs with Hyperadrenocorticism"

_animals, 2023, doi:10.3390/ani13203244_

Round 1

Reviewer 1 Report

Dear Authors, please follow these suggestions:

Lines 122-124: to confirm the diagnosis of PDH, did you perform direct image diagnostics (MRI, CT)? 

Line 147: is it the correct name of the food used?

Table 1: please clarify

Table 2: move the table after the results (and before the discussion)

Lines 221-224: please clarify

Lines 261-262: please clarify

Line 296: did the dogs with HAC receive hepatoprotectants and choleretic also during the study period? This could affect the results

Reviewer 2 Report

Thank you for submitting this study. The idea is interesting but the study is very small and of short duration and most dogs were not controlled at the end of the study. Therefore, I think your conclusions are too strong for a study which is at most of preliminary value. There is some information lacking in Materials and Methods section and the Discussion needs substantial changes in terms of structure and conclusions. Please find more specific comments below.

Introduction

Line 62 – Please replace „is“ by “was” or “used to be”

Line 68 – Please replace “diagnoses” by “diagnosis”

Line 68 – Is this sentence referring to dogs with HAC or dogs generally? Please clarify.

Line 76 – “Serum total bile acids…” – I would put this sentence at a start of a new paragraph.

I am not entirely sure if evaluation of clinical and biochemical parameters were your main aims because the title does not reflect this and you mainly concentrate on the SBA rather than these parameters. Please reconsider your aims/objectives.

Materials and Methods

Please state if you had any exclusion criteria for the HAC dogs. E.g., in limitations you discuss the diet, but you have not stated that you only included dogs on stable diet. You also mention that some dogs were treated with hepatoprotectants… It would be interesting to now how many dogs did not pass inclusion criteria as you obviously had some.

Line 118 – Please replace “arrive at the diagnosis” by “make the diagnosis”

Line 127-128 – Please move this sentence referring to Table 1 on a new paragraph, otherwise it’s a bit confusing. Ideally, use “both groups” instead of “each group” as you only have 2 groups. Generally, this information should actually be included in the Results rather than Material and Method section, but if other reviewers don’t mind, it can stay in Materials and Methods.

Line 134 – Were clinical signs also taken into consideration when making changes to the trilostane dose? What were the dose increments? What cortisol concentration following ACTH was considered good HAC control? Also please move the information about the timing and performing of ACTH stim test up here.

Were dose adjustments made after 4 weeks already?

Line 143 – “Osmolalities”? Maybe “concentrations”??

General question, the healthy dogs presented only once, right?

Please also provide the data for validation of SBA measurement using Catalyst. Is there a publication? How accurate is this machine to measure SBA? How many dogs were used to establish the reference interval?

Results

Lines 172 – 181 -I think these results would be better presented as a bar chart or a table.

Line 192 – It would be interesting to list at this stage in how many HAC dogs the ALT, ALP and cholesterol remained were increased at the end of the study period.

Line 197-198 – “concentrations” instead of “osmolalities” please

Please make sure the order of information in the paragraph about SBA makes sense. Currently is chaotic. See comments below:

Line 203 – “were” instead of “was”please

Line 204 – The information about the lack of difference between healthy and HAC dogs concerning post-prandial SBA should be moved here. Is this meant at treatment start?

In line 204-205 you say that there was no change in pre-prandial SBA during trilostane treatment. What about the posprandial?

Line 210 – You had a number of healthy dogs with abnormal posprandial SBA. Please add this information in results and discuss in the Discussion section. How do you explain this? Were those dogs really healthy?

Line 212-214 – I agree, this is interesting, but actually at month 1 and 3 it is looking more like in the healthy dogs. Maybe elaborate more on this here or in the discussion.

In Table 2, please add the reference range of the laboratory.

Discussion

Generally, the discussion is poorly structured and very difficult to follow. I would suggest starting summarizing the results and then start explaining and discussing the findings in more detail in the following paragraphs. If you claim you are looking at clinical signs, why do you start discussing your final objective (i.e. SBA) first? Doesn’t make sense to me.

Lien 216 – You claim in the Introduction your primary aim is to evaluate clinical signs, secondary to evaluate biochemical results. It is confusing that you say one very generic sentence about clinical signs and biochemical results and then jump straight into SBA…Please give the discussion a short introductory paragraph summarizing the results.

Also it is confusing that you start explaining the cholestasis and then also possibility of pre-prandial SBA being high then post-prandial SBA e.g. due to gall bladder contraction…you have not describe pre-prandial SBA in your dogs being higher than post-prandial in your population or selected dogs…so I am not sure where is this coming from.

Line 254-255 – Is this all still some sort of introductory paragraph? And where is this stamen coming from? Literature? It is not something you could really show by your data. I am confused.

Line 256-259 – This is something which would be better place at the end of your paper. But generally the structure of the start of the discussion is very unclear to me so not sure why do you speak about future studies here.

Lines 261 – 276 – This sounds plausible but how do you explain that some of your healthy dogs also had increased post-prandial SBA?

Line 282-283 – What is meant by notable? Significant?

Line 286 – Is the study referenced here somehow contradictory to studies used to reference the previous sentence? (ref 1,3,37,38) You start the sentence with “however”, so I am wondering why?

Line 288-290 – What is the reference for this statement? Is it based on histology or is it just an assumption?

Regarding this paragraph about ALT and ALP, it would be good to say that this is not really new that HAC dogs have higher ALT and ALP than healthy. This has been shown previously. The same for cholesterol, for example. This should be said.

Line 302 – I don’t really know why your results were different. You could shown reduction of cholesterol on a different trilostane dose. Please rephrase.

Line 312-314 – Please add the reference.

Line 316-317 – I am afraid I do not really understand this conclusion. Please clarify.

Line 320 – The trilostane dose you used is lower than in package insert but I guess recommended by some endocrinologists (there are papers on this), otherwise you would not have chosen this dose. Please elaborate/explain.

You will need to discuss and include as the most important limitation of your study that your dogs were followed over a short period of time, with actually the same trilostane dose at the end than at the start – which is basically due to short duration – which is why these results are purely preliminary. Many of your dogs were not controlled clinically or based on ACTH stim test and therefore you can basically only say that liver enzyme activities and cholesterol did improve but did not normalize in all dogs and the lack of change in SBA might be due to short duration and insufficient control of HAC. You cannot really conclude anything else from your study. It’s difficult to claim SBAs don’t improve because they might improve if you control the disease but we just don’t know from your study. This needs to be clearly stated throughout.

Line 332-333 – Please add a reference for the Na/K ratio as marker for hypoadrenocorticism in dogs on trilostane treatment.

Line 339 – Please replace “osmolalities” by “concentrations”.

Line 338 – 339 – Please rephrase the “sufficiently small magnitude”

Line 352 – I guess you mean dogs with HAC that did not receive any treatment, right? I guess it would not be ethical to withhold treatment, right? You should say that.

Line 354 – What do you mean by “uncontrolled variables”?

Line 355 – How would you assess disease severity in the HAC dogs? You should make some suggestion so that it can be included in future studies. But I am not aware of such tool to assess disease severity in Cushing dogs, to be honest. But I guess you might be as you are discussing it?

Line 365-368 – This is a very strong conclusion for the fact that your dogs were treated for short time period and their disease was not controlled at the end of the study.

Line 368-372 – How would dogs with HAC benefit from the reduction or normalisation of SBA? Is this an important measure of treatment success? Why? Because you even suggest we need to improve treatment protocols to optimize the SBA but I don’t see why is this important clinically.

Some improvement is needed, but generally it is mostly ok.

Round 2

Reviewer 2 Report

Thank you very much for making the changes. I think the manuscript has considerably improved and is ready for publication.

Some mild changes to English language are needed - there are a few missing words but nothing major. Please read through carefully before submitting the final version. Also, I would suggest to create more paragraphs in the Discussion, so that the long paragraphs are a bit easier to follow. But the content is fine, is just to make it a bit easier to read.